# Virtual Disassembling of Historical Edifices: Experiments and Assessments of an Automatic Approach for Classifying Multi-Scalar Point Clouds into Architectural Elements [note 1]

**DOI:** 10.3390/s20082161

**Published:** 2020-04-11

**Authors:** Arnadi Murtiyoso, Pierre Grussenmeyer

**Affiliations:** Photogrammetry and Geomatics Group, ICube Laboratory UMR 7357, INSA Strasbourg, University of Strasbourg, F-67000 Strasbourg, France; arnadi.murtiyoso@insa-strasbourg.fr

**Keywords:** heritage, 3D documentation, point cloud, automation, segmentation, classification, GIS

## Abstract

3D heritage documentation has seen a surge in the past decade due to developments in reality-based 3D recording techniques. Several methods such as photogrammetry and laser scanning are becoming ubiquitous amongst architects, archaeologists, surveyors, and conservators. The main result of these methods is a 3D representation of the object in the form of point clouds. However, a solely geometric point cloud is often insufficient for further analysis, monitoring, and model predicting of the heritage object. The semantic annotation of point clouds remains an interesting research topic since traditionally it requires manual labeling and therefore a lot of time and resources. This paper proposes an automated pipeline to segment and classify multi-scalar point clouds in the case of heritage object. This is done in order to perform multi-level segmentation from the scale of a historical neighborhood up until that of architectural elements, specifically pillars and beams. The proposed workflow involves an algorithmic approach in the form of a toolbox which includes various functions covering the semantic segmentation of large point clouds into smaller, more manageable and semantically labeled clusters. The first part of the workflow will explain the segmentation and semantic labeling of heritage complexes into individual buildings, while a second part will discuss the use of the same toolbox to segment the resulting buildings further into architectural elements. The toolbox was tested on several historical buildings and showed promising results. The ultimate intention of the project is to help the manual point cloud labeling, especially when confronted with the large training data requirements of machine learning-based algorithms.

## 1. Introduction

Documentation of heritage objects by means of surveying techniques has a long history. Indeed, surveying techniques have been an integral part of any conservation effort as well as archaeological missions since the early days of heritage conservation [1]. The need for geospatial data was, and remains, important in order to present a real and tangible archive. While 3D techniques have been used for at least several decades for this purpose, they have seen a very important development since the beginning of the third millennium. This is due to significant improvements in the quality of 3D recording sensors, including the invention of the laser scanning or LIDAR technique [2]. Fast and accurate heritage recording became feasible, although it remained an expensive endeavor. However, during the last decade, further improvements in both hardware and software have rendered the 3D nature of heritage documentation more and more ubiquitous. The term “reality-based 3D modeling” was introduced, which nowadays depend on mainly two methods: passive or image-based and active or range-based [3].

Photogrammetry represents the most commonly used technique in the image-based approach. A branch of science which has a long history in 3D data generation since the advent of aerial photography during the early 20th century [4], photogrammetry has seen massive improvements in terms of computing capability as well as the results offered. The traditionally surveying-oriented photogrammetric process was augmented by various techniques from the computer vision domain, such as Structure from Motion (SfM) and dense matching, to create a versatile and relatively low-cost solution for 3D heritage recording [3]. Of course, improvements in lens and sensor capabilities as well as the democratisation of drones have also improved photogrammetry’s popularity even further amongst the heritage community [5].

As far as the active range-based approach is concerned, the LIDAR technology (including both the Terrestrial Laser Scanning or TLS and the Aerial Laser Scanning or ALS) has also developed tremendously. Using as comparison parameter the scan rate of Time of Flight (ToF) devices produced for example by Trimble, the point per second rate has improved exponentially from 5000 points/second (Trimble GX) in 2005 [6] to 25,000 points/second (Trimble SX10) in 2017 [7] and 100,000 points/second (Trimble X7) in 2020 (https://geospatialx7.trimble.com, retrieved 22 January 2020). This is also supported by significant improvements in the software part, with workflow automation taking more and more importance aided by ever more powerful computing cores available for the user [8,9].

The most common result of the 3D recording process is a 3D point cloud, either via direct laser scanning or the deployment of dense matching algorithms on an oriented photogrammetric network. The point cloud stores geometric information (i.e., XYZ coordinates) which forms a 3D representation of the thence scanned object [10]. Several other pieces of information can also be stored within the point cloud, commonly other geometric features such as point normals, curvatures, linearity, and planarity (relative to a local plane) [11] as well as RGB color or scan intensity in the case of a TLS point cloud. However, this information remains singular for each point within the point cloud. In order to be able to perform more meaningful operations on the point cloud, segmentation must be performed and followed by semantic labeling, thus virtually disassembling the raw point cloud into smaller, classified clusters [12]. The segmented clusters of point clouds can thenceforth be treated as classified point cloud, from whence various analyses, 3D modeling, and model predictions could be performed. Indeed, a special subset of Building Information Model (BIM) is dedicated for heritage buildings, dubbed the Heritage Building Information Model (HBIM) [13], which enables such operations to be conducted.

Nevertheless, even before the 3D modeling for HBIM starts, the process of point cloud segmentation and classification is largely a manual process; an operator manually segments and labels each point cloud cluster as the intended entity. This process is similar and even analogous to traditional digitising in the 2D case of aerial or satellite images into 2D vectors with attributes in GIS (Geographical Information System). The manual segmentation and labeling is further complicated in the case of heritage objects due to its inherent complexity in terms of architectural style, materials, age of the structure, infinitely diverse decorations, etc. It is therefore more often than not the most time consuming part in the 3D pipeline [14] and therefore presents a bottleneck in the general workflow.

This paper intends to present our work on a series of functions (collectively stored as a toolbox) which enables an automatic processing of some parts of the point cloud segmentation and classification problem in the case of heritage objects (see Figure 1). Taking into account the complexity of the problem (mainly due to the differing architectural styles and elements involved), the project addresses only the processing of several particularly important architectural classes, such as structural supports (pillars, piers, etc.) and framework supports (be it wooden or otherwise). The proposed toolbox also works in a multi-scalar approach, meaning that the input point cloud is processed on several levels of scale, from that of a heritage complex up to architectural elements. The flexibility of the toolbox is intended to enable an easier adaptation for different types of heritage sites. Furthermore, several comparisons with other existing approaches will also be presented to assess the reliability of the developed algorithms.

## 2. General State-of-the-Art

The documentation of heritage objects has been addressed in a lot of literature. As has been established beforehand, the documentation process takes more and more the form of 3D recording. Nowadays, the use of image-based (e.g., photogrammetry) and range-based techniques is very common [3,15] and may even be complementary to each other. Several useful guidelines also exist to advise stakeholders who do not have a surveying background on good practices in the subject [10,16]. Numerous examples exist in the literature on the use of 3D techniques for heritage documentation, e.g., the work of [17,18,19] to cite a few. Another trend that has surfaced as a logical consequence of the availability of multiple sensors is data integration, both in the sensor level and the point cloud level [20,21,22,23].

### 2.1. Point Cloud Processing

Several approaches to point cloud processing exist in the literature. A very general division of point cloud segmentation and classification is given in [24], in which the existing algorithms are divided into either the use of geometric axioms and mathematical functions, or the use of machine learning techniques. This division is concurrent with ideas presented by [25], in which the former is mentioned as the use of geometrical, spatial, and contextual constraints. The authors in [26] mentioned a distinction between heuristic and machine learning techniques. Another attempt to classify the existing approaches was proposed by [27], in which the authors added region growing algorithms [28,29], edge-based segmentation [30], and model fitting [31] as other possible segmentation approaches, while point cloud classification is divided into supervised (data-training), unsupervised, or interactive manner.

#### 2.1.1. Machine Learning and Deep Learning Approaches

In general, machine learning and its subset deep learning solutions have seen a surge in popularity in these recent years since the advent of big data [32]. Machine learning approaches are robust against noise and occlusions, and generally reliable. Its main disadvantages, however, is the necessity of a large amount of training data and the computing power needed to train the algorithm. The usual method to create training data is to segment and classify point clouds manually [33], although synthetic training data can also be generated in some cases [34]. It also remains a largely black-box solution and therefore leaves very little room for user intervention [26].

Various types of machine learning and deep learning techniques are available, as described in [32]. In [35], a comparison on several machine learning and deep learning techniques were performed. The authors in [36] described a deep learning approach to classify outdoor point clouds in the case of heritage sites, while the authors in [37] proposed the use of a multi-scalar approach for classifying multi-resolution TLS data. As deep learning is a well-established technique in the realm of 2D image recognition, one way to perform point cloud classification is to apply deep learning on 2D images created from point cloud color (orthophotos, UV textures, etc.) [38]. The technique is also often used to perform the segmentation and classification of point cloud generated by aerial platforms (aerial photogrammetry, ALS) as it enables the reduction of the (usually more complex) 3D point cloud into a 2.5D problem [39].

While the appeal of machine learning is strong for performing point cloud processing in the case of complex geometries as encountered in heritage objects, the main bottleneck remains the generation of the training dataset [25]. In this paper, an algorithmic approach is considered in order to provide a fast result which may eventually be used to help generate training data for future machine learning techniques. Indeed, manual labeling of heritage objects also present a particular difficulty since objects in the same class may have many variations.

#### 2.1.2. Algorithmic Approach

The algorithmic approach employs geometric rules and mathematical functions to perform point cloud segmentation [40]. This approach is often heuristic in nature, but maybe enough for certain purposes as they are fast and simple to implement [26]. Algorithmic segmentation uses mathematical rules and functions as constraints during the segmentation (and possibly also classification). These rules may range from simple rules (e.g., “floors are flat and located below each storey” or “pillars are cylinders”) [25] to the implementation of ontological relations [41,42,43].

The rules and constraints in this type of method are often determined differently according to the encountered case. The authors in [44] employ a type of multi-scalar approach by sub-dividing the point cloud into floors, rooms, and thence walls. The authors in [45] similarly use geometric constraints to segment the walls of bridges. In [41], relational ontology was used as constraints in determining the classes of point clouds segmented by connected component segmentation.

It is worth noting that most of the examples seen in the literature address a particular scale level for the point cloud. For example, the authors in [39] focuse on small scale point clouds or larger areas mainly done to support surveying purposes, while the authors in [26] perform the point cloud processing at the larger scale of a building. Many algorithms were also developed with modern objects in mind [44,45] even though forays into the heritage domain is becoming more and more numerous in recent years [36,38]. The goal of this research is to develop a toolbox which enables the processing of multi-scalar heritage point cloud, from the scale of a neighborhood (heritage complexes) up to that of architectural elements. This is encouraged by the increasing trend towards multi-sensor and multi-scalar recording missions for heritage sites [22,46].

### 2.2. Automation in 3D Modeling

The next stage in the 3D pipeline at the end of point cloud classification is 3D modeling. This step once again presents a bottleneck point in the pipeline. This project will not address 3D modeling automation in detail as it remains a future work; however, some preliminary results in the case of wooden beams will be presented in the paper.

In light of the largely manual operation in 3D modeling [25], its automation has therefore been a very interesting subject of research in recent years. The modeling of planar surfaces or façades has been studied in many works [47,48]. These approaches often employ surface-normal similarity and coplanarity in patches of vectorial surfaces. Another uses robust algorithms such as the Hough transform or RANSAC to detect the surface [30]. Once a planar region is detected, the parameters of the plane can be estimated using total least squares fitting or robust methods that are less affected by points not belonging to the plane [49]. In regard to indoor modeling, the methods are mainly based on geometric primitive segmentation. Some approaches are based on space segmentation before applying a more detailed segmentation [40]. The segmentation of planes is then performed using robust estimators such as MLESAC, which uses the same sampling principle as RANSAC, but the retained solution maximizes the likelihood rather than just the number of inliers [50].

In particular regard towards HBIM, problems related to the complexity of certain heritage buildings exist. In this case, the Level of Detail (LoD) requirements of heritage objects are often higher [14,51]. One of the main challenges is the standardization offered by current BIM technology used to manage simple buildings and constructions [52], limited by the irrelevance of object libraries and the inability of 3D scans to determine structures in buildings of dissimilar age and construction [53]. Indeed, several research [53,54] focused on enhancing the existing libraries of historical parametric objects, but few address the automation in HBIM generation.

## 3. Nature of Available Datasets

The research described in this paper utilizes several datasets which are mainly heritage sites (Figure 2). The main datasets both involve multi-sensor and multi-scalar data. The multi-sensor aspect is due to the fact that the final point cloud is a result of the combination of several 3D sensors, in most cases photogrammetry (aerial and close range drone as well as close range terrestrial photos) and laser scanning (terrestrial, but also aerial LIDAR in the case of the St-Pierre dataset). The multi-scalar aspect is achieved by the recording of not only one particular building of interest, but also the heritage complex or the neighborhood around it. This is done in order to have a complete documentation of the heritage site within the context of its geographical location.

The method of 3D data integration follows the existing workflow as described in [21,46]. In order to create a common coordinate system, each of the available 3D data was georeferenced separately into the same geodetic coordinate system. In order to do this, topographical surveys were conducted in parallel to photogrammetry and laser scanning in both of the main datasets. Artificial targets were measured and thereafter integrated in the absolute orientation phase for photogrammetry and the georeferencing process of a TLS point cloud. The chosen coordinate system corresponds to the respective national mapping projection system of each entity, therefore ensuring that future projects may also be integrated easily.

The multi-scalar aspect of the datasets is directly linked to the multi-sensor aspect. In general, both the main datasets were recorded in different scales in order to have different levels of details; i.e., the neighborhood scale level comprising the heritage complex was recorded using either drone photogrammetry or aerial LIDAR, the building scale level comprising individual heritage building exterior was recorded using TLS and close range photogrammetry, while the interior scale level as well as the architectural elements were scanned using TLS and in some particular cases also photogrammetry.

In addition, two supporting datasets were also used to augment the research and serve as an objective experiment on the developed algorithm’s performance. These datasets are generally specific in nature, and does not possess the multi-sensor and multi-scalar attribute of the two main datasets. They are, however, useful in order to give another perspective and test the capabilities of the algorithm. The two supporting datasets comprise of one dataset dedicated for the pillar detection algorithm (Section 4.2) and another for the beam detection part (Section 4.3).

The main datasets used in this research are as follows:*Kasepuhan Royal Palace, Cirebon, Indonesia* (**“Kasepuhan”**): This historic area dated to the 13th century and includes several historical buildings within its 1200 m^2^ brick-walled perimeters. A particular area of the dataset called Siti Inggil is of particular interest to the conservators as they represent the earliest architectural style in the palace compound. In this paper, the Siti Inggil area is used as a focal point, with one its pavilions (the Central Pavilion) used as a case study for the more detailed scale level. Heavy vegetation was present within Siti Inggil often overlapping with the buildings, which will provide a particular challenge for the algorithm described in Section 4.1. The site was digitized in May 2018 using a combination of TLS and photogrammetry (both terrestrial and drone), and was georeferenced to the Indonesian national projection system.*St-Pierre-le-Jeune Catholic Church, Strasbourg, France* (**“St-Pierre”**): The St-Pierre-le-Jeune Catholic Church was built between 1889 and 1893 in Strasbourg during the German era. The church is located in a UNESCO-listed district, the Neustadt, which comprises some other historical buildings of interest such as the Palais du Rhin, formerly the Imperial palace during the German Reichsland era between 1871 and 1918. It is an example of neo-Romanesque architecture crowned by a 50 m high and 19 m wide dome. The neighborhood around the church was used as a case study in the research along with its interior. The church’s surroundings was scanned by aerial LIDAR in 2016 by the city’s geomatics service; the point cloud data have since been published as open data (https://data.strasbourg.eu/explore/dataset/odata3d_lidar, retrieved 24 January 2020). The exterior of the church was also recorded using drones in May 2016 to get a larger-scale and thus more detailed data, while the interior was scanned using a TLS in April 2017.

The supporting datasets are as follows:*Valentino Castle, Turin, Italy* (**“Valentino”**): The Castle of Valentino is a 17th century edifice located in the city of Turin, Italy. It was used as the royal residence of the House of Savoy and was inscribed into the UNESCO World Heritage list in 1997. Today, the building is used by the architecture department of the Polytechnic University of Turin. The particular “Sala delle Colonne” or Room of Columns inside the castle was used in this study. This point cloud has been graciously shared by the Turin Polytechnic team for our experimental use. The Valentino dataset is used exclusively for the pillar detection part of the research.*Haut-Koenigsbourg Castle, Orschwiller, France* (**“HK-Castle”**): The Haut-Koenigsbourg is a medieval castle (dated to at least the 12th century) located in the Alsace region of France. Badly ruined during the Thirty Years’ War, it underwent a massive, if somewhat controversial, restoration from 1900 to 1908. The resulting reconstruction shows the romantic and nationalistic ideas of the German empire at the time, the sponsors of the restoration. The castle has been listed as a historical monument by the French Ministry of Culture since 1993. In this research, only a part of the timber beam frame structure of the castle scanned using a TLS was used to perform tests on the beam detection algorithm. The beams are mostly oblique and distributed in the 3D space. The beams are of very regular shape and relatively unbroken [55]. The HK-Castle dataset is used exclusively for the beam detection part of the research.

## 4. The M_HERACLES Toolbox

When addressing the 3D documentation of heritage, in many cases, the historical edifice of interest is located within a larger heritage complex or site. A thorough documentation may therefore incorporate this larger area, often at the scale of a neighborhood, into the mission. In this regard, a multi-scalar and multi-sensor approach is unavoidable; each sensor is usually adapted only for one object scale, e.g., close range photogrammetry for statues or phase-based TLS for building interiors. Indeed, smaller scale objects or larger areas do not need the same fine resolution as an artefact or architectural detail. To address this issue, we propose not only a thorough multi-scalar recording of heritage complexes, but also a progressive point cloud processing. The multi-scalar 3D data acquisition pipeline has been adequately explained in [46]. This section will address the multi-scalar point processing part, starting from the scale of a neighborhood up to that of architectural elements.

As previously evoked, in this study, the multi-scalar approach is used to progressively segment the point cloud of a heritage complex—first into building units and then further into architectural elements (e.g., wooden frames and structural supports). The general flow of the approach can be seen in Figure 3. In addition to progressive segmentation, the developed method will also try to classify the results as automatically as possible in order to add the semantic dimension to the data, which is vital in BIM and 3D GIS environments. In this regard, a toolbox was created in the Matlab^©^ environment to host all the codes and functions written for the study under one project: HERitAge by point CLoud procESsing for Matlab^©^ (M_HERACLES). The aim of M_HERACLES is to develop simple algorithms to help in the automation effort of point cloud processing in the context of cultural heritage. This includes among others segmentation, semantic annotation, and 3D primitive generation. The toolbox is open source and available online via GitHub (see Appendix A for download link). M_HERACLES is developed in Matlab^©^ R2018a using its Computer Vision Toolbox and several other third party libraries.

The datasets were all processed using functions available in the M_HERACLES toolbox; the two main datasets (Kasepuhan and St-Pierre) were processed firstly on the scale level of a heritage complex to that of individual buildings (step 1 in Figure 3). The resulting sub-clouds were thereafter used to be further segmented on the building to architectural element scale level (step 2 in Figure 3). More specifically, Kasepuhan, St-Pierre, and the supporting dataset Valentino were tested for pillar detection (Section 4.2) in this step, while the HK-Castle dataset was used to test the beam detection function (Section 4.3) in M_HERACLES. All datasets were processed using an Intel^(R)^ Xeon^(R)^ E5645 2.4 GHz CPU.

### 4.1. Step 1: Using GIS to Aid the Segmentation of Large Area Point Clouds into Clusters of Objects

This section is an extended version of the work previously presented in [56]. Additionally, the St-Pierre dataset was also added to provide another experimental result. An updated statistical analysis will also be presented in this section.

#### 4.1.1. Rationale and Description of the Developed Approach

GIS has been used extensively for heritage site management [57,58,59] as it enables the integration of both geometric and semantic aspects of the object. GIS in this regard is often available in 2D, comprised of vectorial digitizations of overhead objects and their semantic attributes. One of the most widely used formats for GIS is the ESRI shapefile (.shp) format [16,60].

Several approaches exist in the literature as to automating the object segmentation process, including the use of region-growing methods [61,62]. Another possibility presented by [63] computed normals on aerial point clouds and performed an analysis based on a tensor voting scheme to classify between man-made and natural objects. The authors in [64] suggested using GIS to help with this segmentation work, but stopped short from integrating the semantic attribute into the entities. Another inspiration to the developed algorithm is the work of [65] in segmenting 2D aerial images.

The developed algorithm was detailed in our previous publication [56]. The main idea behind the segmentation algorithm was to use currently available GIS data, which are often already annotated with semantic information, in order to help with the segmentation of point cloud. The two-dimensional GIS is also straightforward to create and to implement; indeed, in the absence of a GIS data, a shapefile digitization can nowadays be performed quite easily from digital orthophotos or satellite images. In this regard, the input point cloud for the algorithm may come from any source: photogrammetry, aerial LIDAR, TLS, or any combination of those. The only prerequisite is that the point clouds should be georeferenced in the same system as the GIS data. Following the data integration workflow previously established via georeferencing to a common geodetic system, this prerequisite does not pose a problem.

The algorithm starts by classifying the point cloud between ground and non-ground elements. The Cloth Simulation Filtering (CSF) method [66] was used in this regard. Algorithm 1 displays the pseudocode of the proposed segmentation algorithm used at the aftermath of the ground extraction process, as written in the function shapeseg.m. In essence, the function creates a 2.5D bounding box from the geometry stored in the shapefile data. All point clouds of all altitudes located inside this “cookie cutter”-like bounding box were segmented into a single cluster. From this cluster, a subsequent Euclidean distance-based region growing was performed to separate the main object of interest from any possible noise, including those present due to the overstacking of vertical objects (e.g., buildings and trees). Finally, the semantic attribute fields as stored in the GIS file (Figure 4) were annotated to the segmented cluster, thus transferring the information from 2D to 3D. The remaining point cloud was then used as input in the next iteration of the process, thereby reducing the time for each iteration.
**Algorithm 1:** Semantic segmentation of heritage complexes aided by GIS data
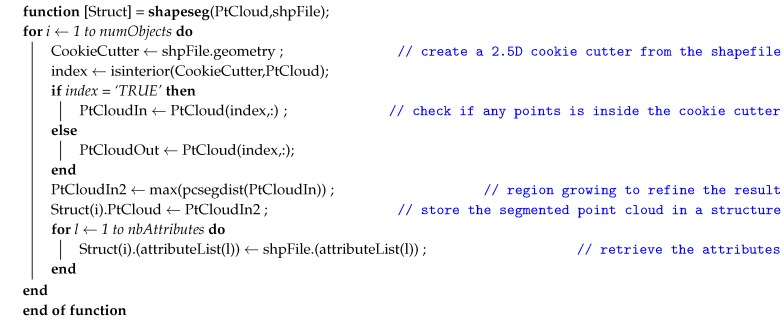


#### 4.1.2. Results and Discussion

The GIS shapefile data shown in Figure 4 were used to aid the segmentation process. In the case of the St-Pierre dataset, the shapefile was acquired through the open data framework of the Strasbourg municipal council, the *“Référentiel topographiques simplifié”* (RTS) or simplified topographic reference. The RTS shapefile data consist of several classes, but, for the purposes of this study, only the “public building” class will be addressed. For the Kasepuhan dataset, however, no prior shapefile was available for the site. The shapefiles of several object classes were therefore generated via digitization of the orthophoto of the site, which was also made available during the acquisition mission. The digitization was made to be not very precise on purpose, in order to test the robustness of the developed function.

The original Kasepuhan dataset consisted of 10.4 million points, and were segmented into four classes (buildings, walls, gates, and the ground) and 13 different annotated objects in about 10 min. In the case of the St-Pierre dataset, the algorithm was visually more successful in segmenting the ALS point cloud into the public building class and annotating them. The St-Pierre dataset consisted of 5.9 million points and was segmented into 17 objects of one class in about 7 min. Here, again, the visually higher success of the St-Pierre dataset may be due to the better CSF results, thus giving a cleaner result than Kasepuhan. Figure 5 shows the results for six of the most important and thus interesting heritage buildings within the class.

Table 1 presents a quantitative analysis on the obtained results for the two datasets. In Table 1, the number of segmented points is used as a parameter of segmentation quality. The “overclassed” column indicates classified points which present a Type I error (false positives), while the “unclassed” column denotes the number of points showing a Type II error (false negatives). Four statistical values were used to assess the quality of the segmentation: percentage of unclassified points (i.e., Type II error rate), precision, recall, and the F1 normalized score.

On the Kasepuhan data, overall, the unclassified rate from all 13 objects yielded an average value of 13.66% and a median value of 6.53%. Meanwhile, the precision median is also quite high at 95.80% with a lower median recall value of 90.65%, thus giving a median F1 score of 91.99%. While this value is seen to be good enough, the quality for each class differs. The buildings and walls class fared the best, with an average F1 score of 93.81% and 94.09%, respectively, although the score for the walls class may be biased since it only consists of two objects. As far as the buildings are concerned, BUILDINGS4 presented the largest error, which is caused by the remaining unfiltered ground around the structure. The walls class presented the worst results with an average F1 score of 84.82%. The WALLS class presents poorer recall value, which may be due to the significant presence of noises. This is particularly true for WALLS5 where the presence of large flower pots rendered the point cloud very noisy. It is also interesting to note that the quality of the ground extraction algorithm played an important role in the results; indeed, worse results were obtained for smaller objects on which the applied ground extraction function fared worse in distinguishing between the object and the ground.

For the St-Pierre dataset, quantitative values seem to validate the qualitative visual inspection of Figure 5 that the algorithm worked better than for Kasepuhan. In the statistical analysis presented in Table 2, only six of the most important heritage sites located within the neighborhood of the St-Pierre church were taken into account. In this reduced sample, the median unclassified percentage amounts to 6.64% (i.e., comparable to that of Kasepuhan), but the median precision attained a value of 98.86% and the median recall that of 93.69%, thus yielding a median F1 score of 93.90%. The best result was obtained for the Palais des Fêtes (PLSFETES) building, with 100% precision (97.5% F1 score). The worst F1 score was obtained by the Palais de Justice (PLSJUSTICE) building. This is due to a chunk of the point cloud of the said building which was visibly not segmented into the cluster. This unsegmented chunk corresponds to the scaffoldings on the building, erected due to renovations. The ALS points of the scaffoldings were so few that M_HERACLES considered them as noise. Apart from this outlier, the most frequently encountered error seems to be related to the presence of vegetation or, analogous to the Kasepuhan data, the minor errors due to prior ground extraction.

An interesting point to note in summarizing these results is the speed of the processing when compared to manual segmentation and labeling. The algorithm, while having several outliers (especially in the presence of important noise), generated good results. This is particularly true in the case of the St-Pierre dataset, where the urban density and particularly flat terrain generated very good results.

#### 4.1.3. Comparison with a Commercial Solution

In order to assess the quality of our developed approach, a comparison was performed with the automatic point cloud classification results of the commercial software Agisoft Metashape. While Metashape is chiefly a photogrammetric software known for its use in image-based reconstruction, it was also recently augmented with a function for multi-class point cloud semantic segmentation. According to the official documentation, Metashape employs a machine learning technique to perform this task; indeed, it asks its users to submit training datasets in order to improve the classification quality in the future. For the purposes of our comparison, Metashape version 1.5.3 build 8469 (release date 24 June 2019) was used.

The Metashape automatic classification was performed on both main datasets: Kasepuhan and St-Pierre. Visual results for the Kasepuhan can be seen in Figure 6. Three classes were defined, namely the ‘buildings’, ‘walls’ and ‘trees’ classes. In Metashape, this corresponds respectively to ‘buildings’, ‘man-made objects’, and ‘high vegetation’ classes. Figure 6 shows that the Metashape automatic classification had difficulties in distinguishing between buildings and walls, with most of the walls classified as buildings. Some parts of the walls were also classified as high vegetation. This may be due to the fact that the Kasepuhan dataset presented a large-scale and thus more complex scene not entirely suitable for the machine learning-trained function. Unfortunately there is no way to verify this hypothesis since Metashape understandably does not divulge their machine learning method in detail. On the contrary, M_HERACLES managed to classify the objects fairly well thanks to the use of shapefiles to guide it. Visually, Figure 6c also showed that some of parts were nevertheless unclassified, notably the walls at the back of the dataset. This may be due to the low resolution of the point cloud for this part of the site (note the same observation on Metashape results).

Table 3 displays a quantitative comparison of the two tested algorithms for Kasepuhan, also visualized via histograms in Figure 7. M_HERACLES managed to outperform Metashape in most cases (yielding a slightly lower F1 score in the trees class), especially in the walls class. The median value of the F1 score for M_HERACLES was 85.30% compared to Metashape’s 60.52%. It showed a lower recall value and higher precision, which may be explained by the fact that the use of shapefiles disproportionately increased Type I error.

When implemented on the St-Pierre dataset, M_HERACLES notably still performed better than Metashape as can be consulted in Figure 8. Metashape produced a very high precision rate; however, this must be understood with a caveat. Indeed, Metashape performed automatic segmentation on all buildings on the scene, whereas M_HERACLES only performed one on the “public building” class as dictated by the related shapefile. This distinction between public buildings and other buildings follows the official categorisation as set by the Strasbourg city geomatics service. In this regard, the results of Metashape were therefore manually segmented to include only the so-called public buildings, thus yielding a slight bias towards higher precision. However, as far as the recall value is concerned, M_HERACLES again outperformed Metashape. This is mainly due to the mis-classification rate of Metashape as can be seen in Figure 9. For example, in Figure 9a, much of the St-Pierre church dome and church towers were misclassified as high vegetation. This played a large role in explaining the low recall value for Metashape. Overall, in terms of F1 score, M_HERACLES also managed to outperform Metashape in this case of highly urban scene as opposed to Kasepuhan’s more closed and isolated complex situation.

As can be seen in this section, the proposed M_HERACLES algorithm managed to perform the classification of point clouds in the neighborhood scale fairly well. Comparison with Metashape also showed that our solution presented very promising and interesting results. Another advantage of M_HERACLES is the possibility to retrieve individual objects instead of a single cluster comprising all of the instances in the same class. This is useful when working with heritage sites, since, in many cases and for various reasons, the user may wish to acquire the point cloud of one or more specific buildings. Furthermore, the possibility to annotate these individual buildings with semantic information derived from the GIS shapefiles also presented an advantage for the developed algorithms. As far as the processing time is concerned, Metashape clocked a much faster time at around five minutes for both Kasepuhan and St-Pierre. This also shows the necessity for further optimization of M_HERACLES in terms of processing time, although the current time is already quite satisfactory considering the results obtained.

### 4.2. Step 2 (1): Automatic Segmentation and Classification of Structural Supports

Please note that this section corresponds to the work previously presented in a conference paper [67]. A slightly updated version of the algorithm will be presented, while new statistical analytics were also added to the discussion on the results. New datasets were also tested in this paper to further validate the developed algorithm.

#### 4.2.1. Rationale and Algorithm Description

Pillars or structural supports in a historical setting are often interesting architectural elements, since they showcase both the engineering know-how and the architectural taste of the builders It is with this reason in mind that the first function was developed to segment structural supports automatically. Additionally, simple geometric rules were implemented in order to be able to identify a column from other types of structural supports. Indeed, this kind of development has been addressed before in the scope of simple pillars, often in an industrial setting [45,68]. In the field of heritage, pillars or supports can be very variable depending on the architectural style and geographical situation, hence making this operation more difficult. Some authors solved this problem by creating a dedicated library of parametric objects [12], while the most common solution remains a manual segmentation [33].

In this paper, geometric characteristics (also called hard-coded knowledge, as exemplified in [25]) were used to help identify the class of the segmented point cloud cluster. In particular, the circular cross-section characteristic of most columns will be used as the main rule in determining if a segmented point cloud is column or not. This approach has been used in several other research works, for example [44], for the creation of as-built BIM elements or [45] for engineering purposes. The authors in [68] also developed a similar approach to the one presented in this paper, albeit implemented for modern columns and without semantic labeling.

The proposed method was described in detail in a previous publication [67]. The main idea behind the implementation of the algorithm includes a preliminary segmentation of the building body from its attic part. This part was done automatically by comparing the surface area of horizontal cross-sections of the building; a significant shift in the cross-section’s bounding box’s surface means that the limit between the body and the attic has been attained.

In the main Algorithm 2, the input of the algorithm is the building’s body as previously segmented, whether automatically or manually. The function then operates the following steps:The function first creates slices of horizontal cross-sections of this point cloud, whereas the middle slice was taken. In this regard, the 3D problem was reduced to a 2D problem; a similar approach was undertaken in [44].A Euclidean-distance based region growing is performed on this middle slice, thereby creating “islands” of candidate pillars.A point cloud filtering is performed using the convex hull area criterion to distinguish the “islands” into potential pillars, walls, or noise.From the list of potential pillars, a further division was made between “columns” and “non-columns”, depending on the circularity of its cross-section. A circular cross-section was classified as potential columns, while the rest are identified as non-columns.A “cookie-cutter”-like method similar to the one explained in Section 4.1 is then implemented using the cross-section of each candidate pillar to segment the 3D point cloud. All points located within the buffer area are considered as part of the entity.In the aftermath of the cookie-cutter segmentation, some horizontally planar elements such as floors and/or clings might still linger in the cluster; a RANSAC plane fitting function was therefore implemented to identify these horizontal planes and suppress them.A final distance-based region growing was performed to eliminate any remaining noise. Thus, the output of the function is a structure of point cloud clusters, labeled as columns or non-columns.
**Algorithm 2:** Pillar segmentation and classification
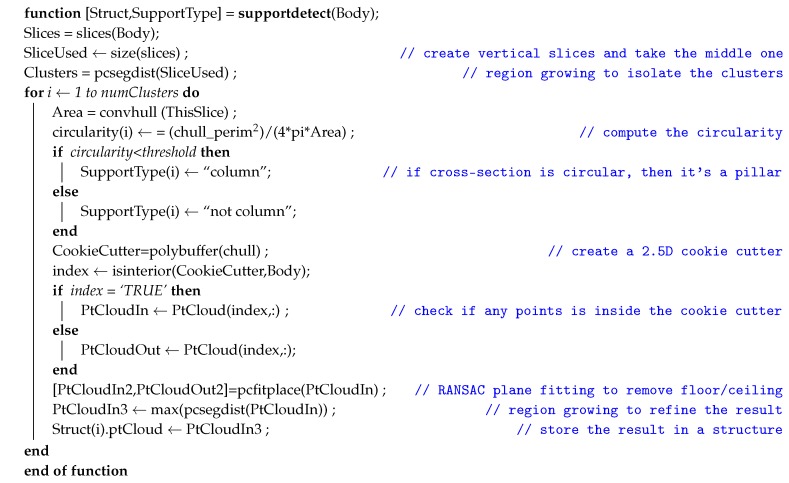


#### 4.2.2. Results and Analysis

A substantial test regarding the results of this section was presented in [67]. In this paper, two new datasets were introduced: the Central Pavilion of the Kasepuhan dataset and the Valentino dataset, while the St-Pierre dataset was presented with new statistics. The results can be consulted in Figure 10. The Kasepuhan dataset is small in size, with a little over 333K points while the St-Pierre and Valentino datasets presented a much larger point cloud with over 1.8M and 3.5M points, respectively. It is also interesting to note that the three tested datasets possess different styles of architecture; the Kasepuhan dataset consists mainly of open pavilions with many free-standing columns while the Valentino presents an example of an interior point cloud case. The St-Pierre church choir was chosen due to its particularity in that it possesses twin pillars instead of the usual free-standing ones.

As can be seen in Figure 10, in terms of the segmentation process, the algorithm managed to detect eight structural supports in the Kasepuhan dataset. This corresponds exactly with the ground truth data. In the case of the St-Pierre, it managed to correctly detect the eight pillars individually despite the twin pillar nature. For the Valentino, 20 structures were detected in lieu of the actual 19 as can be found in the ground truth. The Valentino data presented a particular challenge since 13 out of the 19 pillars present in the dataset are in the form of engaged pillars, i.e., semi-pillars or columns which are part of the wall. As can be seen from the results, the algorithm had difficulties in segmenting these kinds of structural supports, while having no problem for free-standing pillars.

For the Kasepuhan dataset, a preliminary segmentation was performed to divide it into the building body and attic, with the body used as input for the function supportdetect.m as described in Algorithm 2. Some points mainly at the top of the pillars remained unclassified. This is due to the fact that, in the pre-segmentation of the building body and attic, the algorithm considered the change in the surface of the cross-sections of the building to determine the two parts. These cross-section surfaces being calculated from the surface of the bounding-box, only the exterior point cloud was considered instead of the interior. This is reflected numerically in Table 4, where the recall value for this dataset is visibly low despite a very high precision.

The inverse is seen in the Valentino dataset, where the non-planar ceiling created a case of oversegmentation. Indeed, the statistics as for Valentino show a high rate of recall but lower precision. The visual results for the St-Pierre dataset had been amply described in [67], including the overclassification of the iron fence attached to the posterior pillar.

Statistically speaking, within Table 4, Table 5 and Table 6, the overclassified column describes the number of points considered as false positives, while the unclassified column denotes the true negative points. False negative points are not shown since the values are negligible due to the cookie-cutter approach of taking all points of all elevations of a particular polygon shape. Similar to the analysis conducted in Section 4.2, four statistical values were presented to assess the quality of the algorithm, namely percentage of unclassified points, precision, recall, and F1 score. In terms of the unclassed percentage, Kasepuhan showed a higher rate (median of 34.36%) which is most probably caused by the same reasons as the one established before regarding errors during the pre-segmentation between the body and the attic. The median precision of Kasepuhan is 100%, which is very satisfactory. However, as has been previously mentioned, its recall value is lower at 65.64%. This loss in recall value also seems to be systematic, again validating the points as argued in the previous paragraph. The overall median F1 score for the Kasepuhan dataset was 79.23%.

The statistics for the St-Pierre dataset displayed a similar trend to that of the Kasepuhan; that is, it registered higher precision than a recall rate. With a median precision of 81.20% and recall of 71.39%, the results for this dataset are nevertheless quite promising. It should well be noted that the St-Pierre choir dataset is quite complex due to the existence of the twin pillars and the presence of many noises (folded chairs were placed against the twin pillars in addition to the presence of the iron fence on the posterior pillar). Indeed, manual segmentation and labeling took quite some time to perform the task due to these conditions. Granted, the automatic results still had remaining noises and must be cleaned further manually. However, with a fast processing time (a little under one and a half minute), this solution may prove to be very useful in performing the segmentation task, or at least provide a first approximate result.

For the Valentino dataset, the unclassed rate stands at a median value of 11.84%. The precision level is low for the Valentino, at 66.68%, which suggested overclassification. As has been mentioned before, this is mainly due to the ceilings of the dataset that represent arcs instead of planar surfaces as had been hard-coded in the algorithm. A further improvement of the algorithm may incorporate this possibility into account, as this type of ceiling can be found in many heritage datasets. The recall value is, however, quite high, with a median value of 88.16% and thus yielding an F1 score of 75.92%. This means that the algorithm does nevertheless give promising results. Indeed, in some applications where high precision is not necessarily required (e.g., training data generation for deep learning techniques), these results may be sufficient.

As far as the classification goes, Figure 10 shows that the algorithm managed to correctly classify the pillars of the Kasepuhan dataset as non-columns. Indeed, under the definition of columns as set in this paper and contrary to classical columns, the eight structures in the Kasepuhan dataset cannot be classified as columns as they are in fact rectangular shaped posts. For the St-Pierre, the algorithm correctly determined that the eight detected structures are inside the “column” class. In the case of the Valentino, it also managed to correctly identify that the free-standing pillars are columns, while the rest of the detected structures were classified as non-columns.

The processing time of the datasets showed that they may be, at least in part, linked to the number of points inside the input data. However, it is more probable that the bulk of the processing time is linked to the number of detected elements. For the 333K points Kasepuhan data, the algorithm managed to detect, segment, and classify the objects in 25.1 s. This was done in 83.56 s for the 1.8M points St-Pierre dataset, also with eight detected structures. Conversely, the Valentino dataset that consists of almost 10 times more points than Kasepuhan was processed in a little over five minutes in order to detect 20 structures. However, the overall processing time is still faster by at least a factor of 2 when roughly compared to the time it takes to perform the same task manually, without taking into account the time required to identify and classify each cluster into the appropriate classes.

A quick comparison was also performed between our results and the results presented in [36] which also used the Valentino in their experiments with the PointNet++ DL approach. As has been previously mentioned, M_HERACLES managed to yield a median precision value of 66.68%, recall value of 88.16%, and F1 score of 75.92%. In [36], Valentino was used as the test dataset after the authors’ DL algorithm was trained using another dataset and was classified into four classes, including columns. For the columns’ class, the authors cited a value of 49.10% in precision, 70.02% in recall, and 57.60% in F1 score (Figure 11). Although our algorithm managed to provide better results than the compared study, several remarks should nevertheless be taken into account. Firstly, in our study, only free-standing pillars were accounted for, whereas [36] also included engaged columns. Indeed, M_HERACLES did not manage to correctly detect the engaged columns. Secondly, the DL approach used in the other study has the potential to generate better results with more training data.

### 4.3. From Edifices to Architectural Elements (2): Automatic Segmentation of Building Framework

This section describes an ongoing work on the automatic detection of beams in building frameworks. The rationale of this research path is the importance of building frames in the context of historical buildings, as they encapsulate the core of the construction knowledge and know-how of the builders [69]. The recent burning of Notre-Dame de Paris cathedral in April 2019 also emphasized importance in the documentation of the timber framework of other similar structures [70].

The automatic parametric modeling of wooden beams has been addressed in another research conducted by our group, as presented in [55]. However, in that research, the authors relied on total station measurements to automatically create parametric models of wooden beams. The idea between this particular part of M_HERACLES is to benefit from the availability of point cloud data, which is much faster to acquire and provides more data than traditional total station measurements.

Although one might even argue for over-abundance of data in point clouds, the ease of acquisition of points clouds compared to traditional surveying is undeniable. Another similar work of automatic parametric modeling of wooden beams was presented in [71,72]; indeed, the algorithm described in this section took inspiration from their approach.

Contrary to the developments in Section 4.1, Section 4.2 and Section 4.3 where a 2.5D approach was taken, wooden beams present a truly 3D environment where a 2.5D approach was insufficient to solve. The algorithm described in this section therefore takes a departure from the previous lines of reasoning by considering the problem as a 3D one, while still taking notes from the previous algorithms. The idea behind the developed function is to first decompose the beam point clouds into facets. Afterwards, several geometric constraints were applied to extract the point cloud of individual beams from those of the facets.

The functions were created to reach this point of the segmentation process. However, an optional third party library also enables the creating of parametric best-fit cuboids from the segmented beams. The overall workflow of the developed approach is described in Figure 12. The facet detection was performed using the region growing method. The theoretical primer of the region growing method is well known, and, in this case, we used the same approach as the Point Cloud Library (PCL) [73] but implemented it in Matlab^©^. This implementation employs point cloud normals and curvatures as constraints, as opposed to the function pcsegdist in Matlab^©^ which uses Euclidean distance as the principal constraint.

The use of normals and curvature as constraints is important in order to distinguish the different facets. However, the use of a greedy region growing algorithm, implemented in all points, takes too much resource and computing time and is therefore impractical. A solution to this problem is a slight tweak in the algorithm to perform the region growing on octree bins instead of the points themselves [29]. A similar implementation of this algorithm was the fast-marching approach described in [74]. From our observations, this octree-based region growing has shown to increase the computing time up to a factor of 10.

The post-segmented point cloud creates clusters of facets. However, in the case of L or Y-shaped facets, additional segmentation was necessary in order to segment the faces properly for each beam. In our approach, the facet cluster was projected into a 2D binary image via PCA (Principal Component Analysis) transformation, an approach similar to [44,72]. Afterwards, a Hough Transform analysis was performed on the binary image in order to detect the edges. The computed edges for each beam facet were thereafter averaged to obtain the centre axis for each beam facet. When the axis is detected, the L or Y-shaped facet was segmented into individual elongated or I-shaped clusters.

Once the individual facets of individual beams are detected, two geometrical constraints were applied to group the facets into clusters of beams. The two constraints were similar to the ones used in [72], although in this algorithm only two out of the three mentioned in that paper was used. This reduction in geometric constraints was done in order to prevent over-constraining the problem. The two constraints applied in the algorithm are as follows:*Adjacency constraint*: the neighborhood or adjacency constraint was enforced to limit candidate facets of each beam to only facet clusters which are located adjacent to the current facet reference. In [72], this constraint was defined by the distance between the facet centroids. In M_HERACLES, we modified this approach by performing another octree-based region growing on the facets, this time around by enforcing a distance threshold between adjacent octrees from different facets. In this way, adjacency is defined by whether any edge of the facet cluster is near another one.*Parallelism constraint*: once the adjacency between the different facets is defined (via an adjacency matrix), the search for candidate beam facets is reduced to neighbors. Between neighbors, another geometric constraint on the parallelism of clusters was enforced. Firstly, the major principal axis of the facet clusters was computed using PCA. Two facets are considered parallel if their first PCA components satisfy Equation (Equation 1):
(1)$$\vec{OA_{1}}-\vec{OA_{2}}\approx 0$$
where $\vec{OA_{1}}$ is the first PCA component of the first (or reference) facet cluster, and $\vec{OA_{2}}$ the analogous vector for the second (or tested) facet cluster. Since the first adjacency constraint already limited the candidate facet clusters for a beam, this second geometric constraint was deemed enough to detect the beam.

Using these two constraints, the function was able to group the facets into beams. An optional further processing involves using the RANSAC algorithm to generate a best-fit cuboid for each beam cluster. This was however done by a third-party Matlab^©^ library (https://fr.mathworks.com/matlabcentral/fileexchange/65168-cuboid-fit-ransac, retrieved 28 January 2020).

This part of the algorithm is still under development; however, a preliminary result (as shown in Table 7) conducted on a subset of the HK-Castle dataset showed that the algorithm managed to correctly identify the individual beams. The small dataset consists of 100k points and was processed in 3 min 42 s. The algorithm gave very good results in terms of precision (median value of 94.45%), but quite low values of recall (median value of 75.58%). The low recall value can be explained by the fact that the algorithm also performs noise reduction, in which detected regions having less than a set threshold number of points are eliminated. The resulting cluster is therefore cleaner than the manual segmentation, but this means a sharp decrease in recall value. The precision value is however very satisfactory. Furthermore, the algorithm correctly deduced the number of beams that are present in the input point cloud. However, the algorithm still suffers in terms of processing time. More than half of the processing time was taken by the curvature computation at the beginning of the function; this requires therefore more investigation and optimization. Furthermore, these preliminary results concern only a small dataset. More investigations must be conducted to assess the quality of the algorithm, namely by processing a larger dataset.

## 5. Conclusions and Future Work

This paper presents a toolbox of functions dedicated to point cloud processing in the context of heritage objects. The main driving cause of the development of this toolbox is to address the increasingly multi-sensor and multi-scalar nature of heritage documentation. The presented M_HERACLES toolbox enables the user to automate some of the bottlenecks in the 3D processing pipeline of a multi-scalar point cloud, especially in the segmentation of individual buildings from the point cloud of the neighborhood and the detection of two classes of architectural elements. This was done to reduce human intervention and thus human error. Results for the three presented particular functions look promising.

The historical complex to historical building segmentation and classification elaborated in Section 4.1 managed to perform the task correctly, all while retaining the classification according to the input shapefile. The algorithm presented in the paper also managed to automatically annotate the GIS attributes into the segmented clusters in an acceptable processing time, which may prove very useful not only for heritage purposes but also for general mapping purposes. Several caveats still exist, however. As has been previously discussed and detailed in [56], the correct segmentation ordering is important in order to get good results, especially in cases where vertical stacks are present (e.g., trees and building roofs). As a rule of thumb, lower entities (ground, low vegetations, etc.) should be segmented first before taller entities (e.g., buildings, tall vegetation, etc.). As has been previously explained, the ground extraction at the beginning of the algorithm is also an important factor influencing the final product. That being said, the attained median F1 scores of 91.99% for Kasepuhan and 93.90% for St-Pierre are very encouraging.

Comparison with an external solution (Agisoft Metashape) was also performed in this section. Results on both the Kasepuhan and St-Pierre datasets showed that M_HERACLES managed to outperform Metashape as regards the precision and quality of the classification process. However, it should be noted that Metashape uses a machine learning approach in which the availability of training data are paramount. We are quite confident that machine learning solutions will become better as time goes; indeed, one of the main objectives of M_HERACLES is not to confront its performance directly with machine learning solutions, but rather to complement it via e.g., automation of training data creation.

In Section 4.2, tests on two additional datasets in addition to the results described in [67] showed that the algorithm is useful in performing fast segmentation and classification for structural supports. However, as has been shown in the results, the algorithm, while fast and easy to use, remains prone to noise and deviations from the hard-coded geometrical rules. This is evidenced by the stark contrast between the three datasets, whereas Kasepuhan presented cases with higher precision but lower recall (suggesting underclassification), the Valentino showing higher recall but lower precision (suggesting overclassification) and the St-Pierre presenting somewhat of a mix between the two cases. These differences were caused by deviations from the general rule; the Kasepuhan interior ceiling does not correspond to the altitude of the roofs on the exterior, while Valentino’s non-planar ceiling caused the error. The St-Pierre case showed that the proneness of the algorithm against noises. However, the results remain promising and some lessons can be learned from this experiment, to be the subject of further improvement. A comparison with the DL approach presented in [36] for the Valentino also showed a rather favorable outcome with respect to the quality parameters. Furthermore, the fast nature of the segmentation and classification process is in contrast to training-intensive ML/DL and resource-intensive manual segmentation and labeling. It may therefore be used to complement ML/DL algorithms, especially in the generation of training data.

The beam detection algorithm described in Section 4.3 is still in its early stages, and more tests must be conducted in order to assess its efficacy. A test with a small dataset yielded a very satisfactory median precision value of 94.45%, but with low median recall value of 75.58%, which is mostly due to the noise filtering function applied in the algorithm. Although these preliminary results are promising, processing time remains an important issue. Most of the processing time was used for the computation of the normals and especially curvatures. More research should be conducted to optimize this part of the algorithm better.

Many improvements can be envisaged for this toolbox. For example, the use of CAD files in lieu of shapefiles for the Step 1 segmentation part could be very useful especially for some heritage sites where GIS is not available. Being a common file format in the domain of architecture, the CAD file can be another alternative for the segmentation aid. Another idea is to use CAD files to perform a similar algorithm as described in Algorithm 1, but applied to indoor scenes [75]. Another interesting idea which has been planned to be tested is to use the results from Step 2, whether structural supports or beams, to help generate training data for machine learning and deep learning techniques. As has been previously established, one of the bottlenecks in ML/DL approaches is the creation and labeling of training data which is performed manually. The algorithms proposed in this paper may help to automate (or at least provide an “approximate value”) this training data generation process, thus rendering the overall 3D processing pipeline more automatic.

Lastly, the availability of open point cloud data for heritage sites remains difficult due to various reasons. This applies both to ML/DL techniques (in the sense that it reduces the possible training data candidate) and to algorithmic approaches such as M_HERACLES (in the sense that it limits the available data for testing). The creation of an open data portal for heritage point cloud is therefore one of the intended future objective of the research.

## Figures and Tables

**Figure 1 sensors-20-02161-f001:**
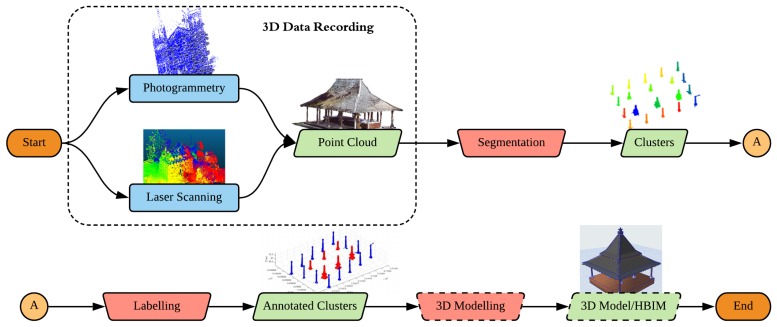
The overall 3D pipeline for the 3D reconstruction of historical edifices, from a point cloud up to HBIM (Heritage Building Information Model)-compatible 3D models. This paper will focus on the manual bottlenecks of the pipeline (red inverted trapeziums) up to before the 3D modeling process (long-dashed elements), although a preliminary result of automatic 3D modeling of beam structures will also be briefly presented.

**Figure 2 sensors-20-02161-f002:**
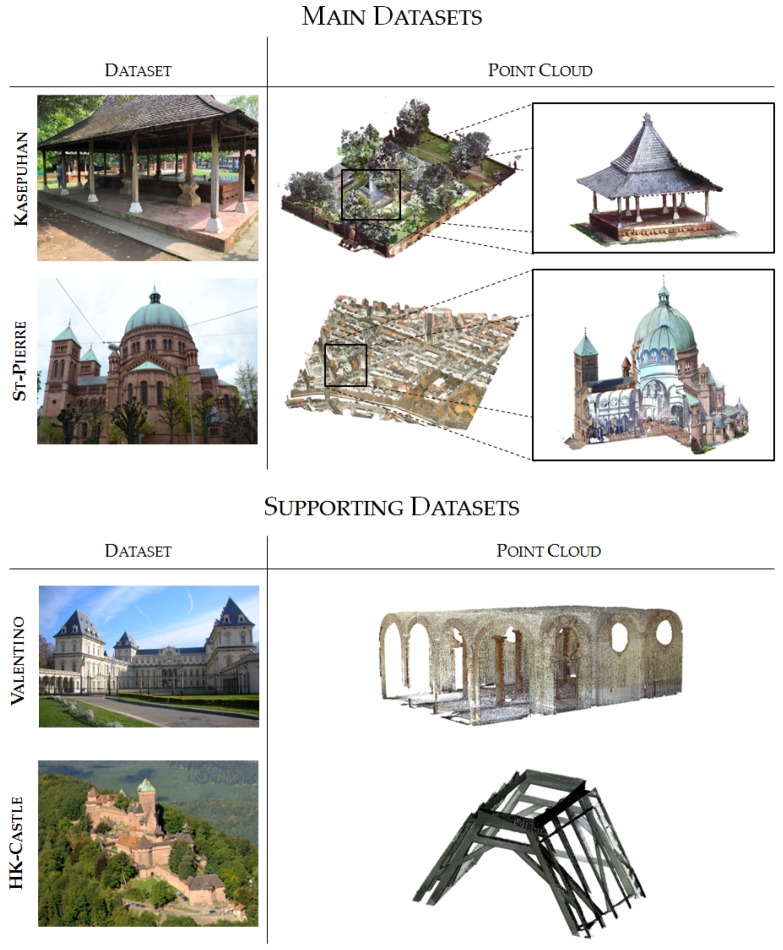
The four datasets used in this research. For the main datasets, the algorithm started with the neighborhood to building segmentation and followed by larger-scale segmentation of an example building (point cloud in the subset) into architectural elements.

**Figure 3 sensors-20-02161-f003:**
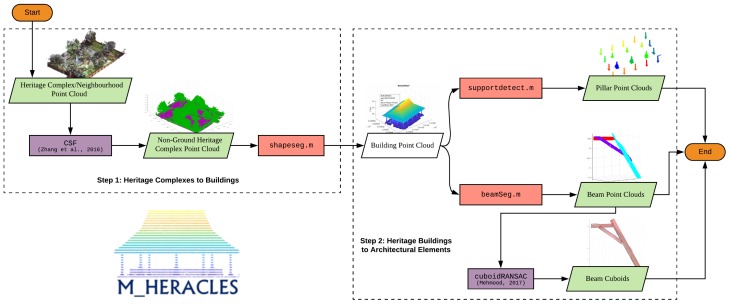
A general flowchart of the workflow within M_HERACLES. Step 1 consists of segmentation from the scale level of a neighborhood to that of individual buildings, while the Step 2 involves segmentation from a building’s scale level to that of architectural elements (pillars and beams). Violet rectangles denote the use of third party libraries; red rectangles denote the main functions developed in this study for the respective tasks.

**Figure 4 sensors-20-02161-f004:**
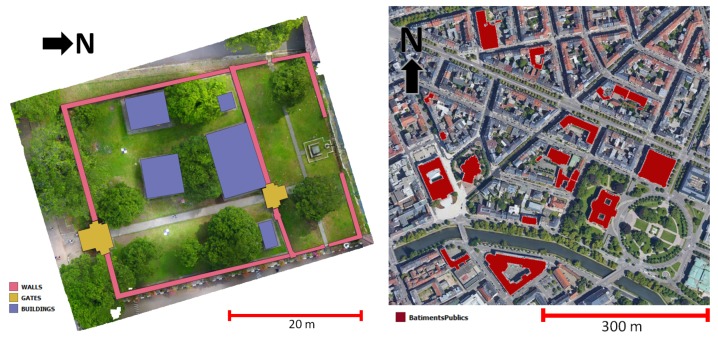
The GIS shapefile data used to help the segmentation process. On the left, three shapefiles were available for the Kasepuhan dataset while, on the right, only one shapefile entity was used for the St-Pierre dataset.

**Figure 5 sensors-20-02161-f005:**
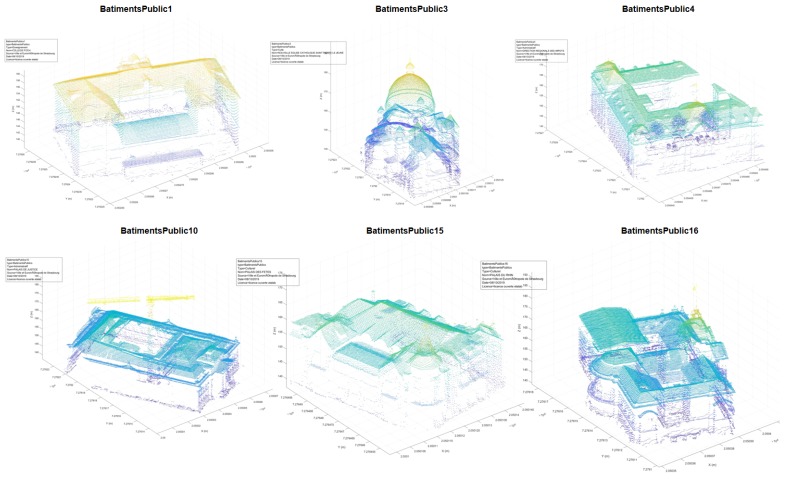
Results of the automatic segmentation and annotation (step 1 of M_HERACLES) for the St-Pierre dataset. Only six of the most important buildings are shown in this figure, of which BatimentsPublic3 represents the St-Pierre church which will be further processed. Note that only the aerial LIDAR data are shown.

**Figure 6 sensors-20-02161-f006:**
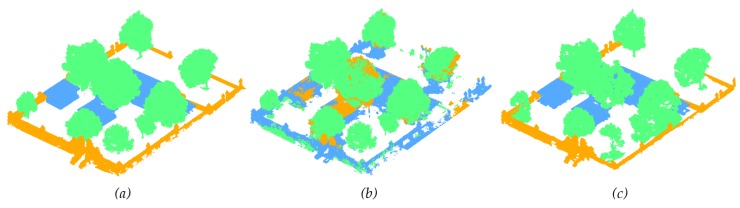
Visual comparison of the point cloud segmentation process: (**a**) displays the manual classification used as reference, (**b**) the results of the Metashape automatic classification, and (**c**) the results from M_HERACLES. The color blue denotes the ‘buildings’ class, orange the ‘walls’ class, and green the ‘trees’ class.

**Figure 7 sensors-20-02161-f007:**
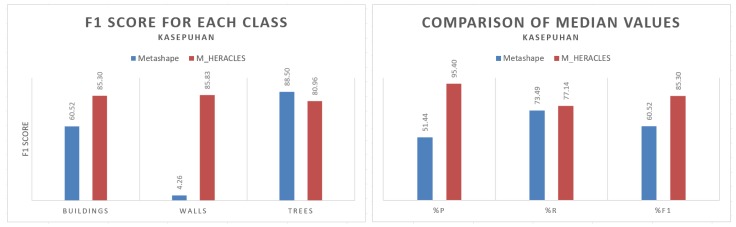
Histogram representation of the classification performance for each class in the Kasepuhan dataset (**left**) and the median value of the principal quality parameters (**right**).

**Figure 8 sensors-20-02161-f008:**
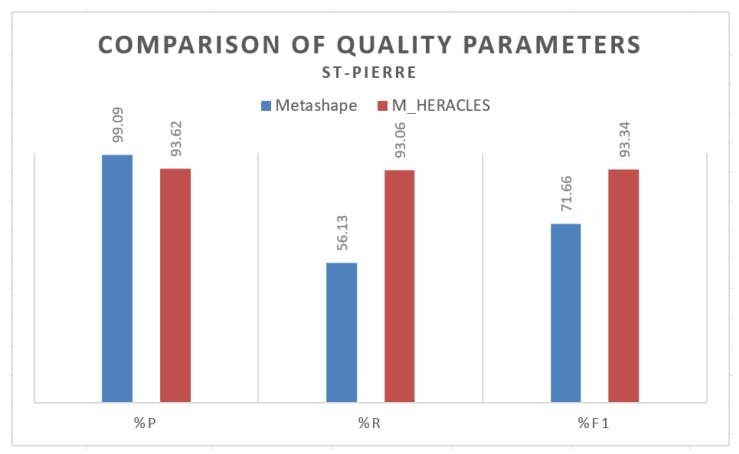
Histogram representation of the quality parameters for the St-Pierre dataset comparing Metashape against M_HERACLES.

**Figure 9 sensors-20-02161-f009:**
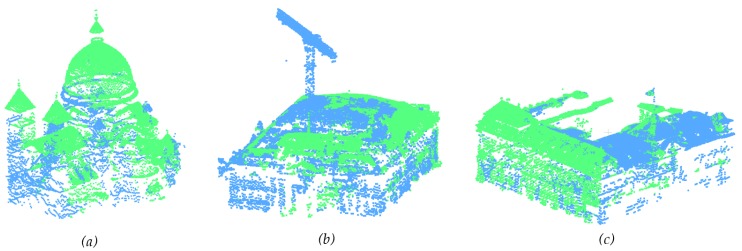
Case of mis-classification in the results of the Metashape point cloud classification on the St-Pierre dataset. Here, some examples are shown where the problem is most observable: (**a**) STPIERRE, (**b**) PLSJUSTICE and (**c**) DIRIMPOTS sub-clouds. The color blue denotes the ‘buildings’ class and green the ‘high vegetation’ class. Note also the presence of the crane in (**b**).

**Figure 10 sensors-20-02161-f010:**
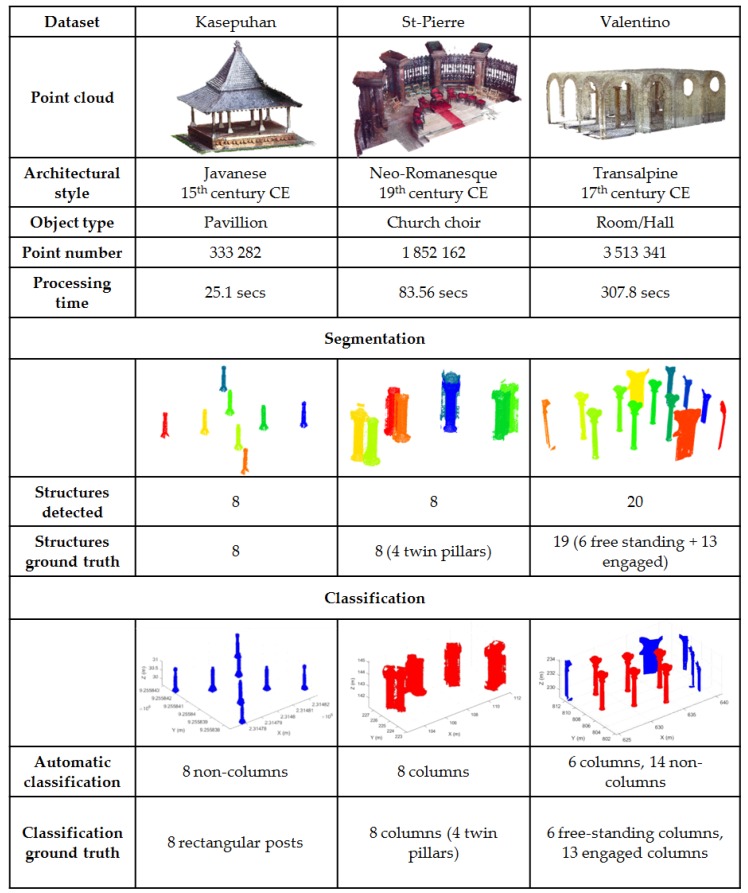
Results of the detection and segmentation of structural supports, as well as their subsequent automatic classification. In the segmentation part, each color denotes a different point cloud cluster, while, in the classification part, red clusters are columns and blue clusters are non-columns.

**Figure 11 sensors-20-02161-f011:**
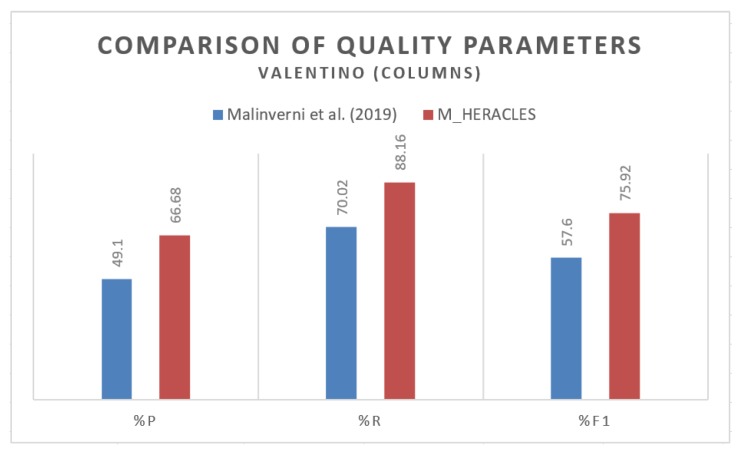
Histogram representation of the quality parameters for the Valentino dataset comparing the results of [36] against M_HERACLES for the column class.

**Figure 12 sensors-20-02161-f012:**
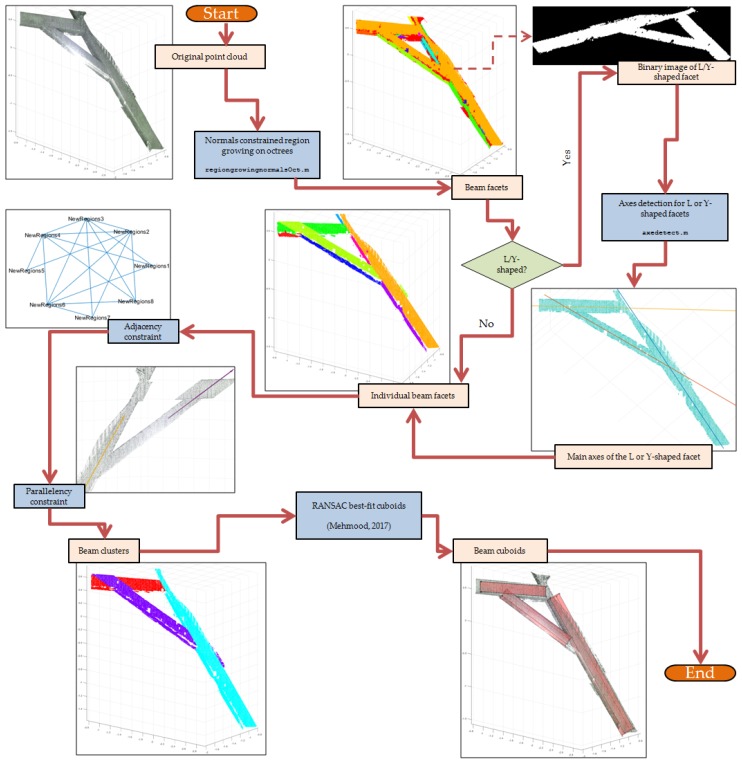
A flowchart of the main steps in the beamdetect.m function of M_HERACLES showing the intermediate and final results.

**Table 1 sensors-20-02161-t001:** The quantitative analysis on the results of step 1 for Kasepuhan. In this table, three classes were taken into account (buildings, gates, and walls) with a total of 13 objects. %P is precision, %R is recall, and %F1 is the normalized F1 score.

Object	Point Number		Misclassed		True Positive	%Unclassed	%P	%R	%F1
	Manual	Auto	Overclassed	Unclassed					
BUILDINGS1	703 500	680 386	10 592	33 706	669 794	4.79	98.44	95.21	96.80
BUILDINGS2	643 350	633 897	6 630	16 083	627 267	2.50	98.95	97.50	98.22
BUILDINGS3	317 459	300 283	9 873	27 049	290 410	8.52	96.71	91.48	94.02
BUILDINGS4	58 532	60 838	8 296	5 990	52 542	10.23	86.36	89.77	88.03
BUILDINGS5	52 026	58 047	7 415	1 394	50 632	2.68	87.23	97.32	92.00
GATES1	101 196	95 754	4 017	9 459	91 737	9.35	95.80	90.65	93.16
GATES2	151 040	146 133	4 955	9 862	141 178	6.53	96.61	93.47	95.01
WALLS1	216 951	151 520	683	66 114	150 837	30.47	99.55	69.53	81.87
WALLS2	417 768	351 818	3 168	69 118	348 650	16.54	99.10	83.46	90.61
WALLS3	84 516	81 520	5 762	8 758	75 758	10.36	92.93	89.64	91.25
WALLS4	64 877	56 804	4 595	12 668	52 209	19.53	91.91	80.47	85.81
WALLS5	63 014	34 752	1 814	30 076	32 938	47.73	94.78	52.27	67.38
WALLS6	177 399	175 862	13 371	14 908	162 491	8.40	92.40	91.60	91.99
					**Mean**	**13.66**	**94.68**	**86.34**	**89.71**
					**Median**	**6.53**	**95.80**	**90.65**	**91.99**

**Table 2 sensors-20-02161-t002:** The quantitative analysis on the results of step 1 for St-Pierre. In this table, only one class was taken into account (public buildings) with a total of 6 out of 17 objects used in the statistical analysis. %P is precision, %R is recall, and %F1 is the normalized F1 score.

Object	Point Number		Misclassed		True Positive	%Unclassed	%P	%R	%F1
	Manual	Auto	Overclassed	Unclassed					
COLLFOCH	34 011	32 384	217	1 844	32 167	5.69	99.33	94.58	96.90
STPIERRE	36 858	34 960	757	2 655	34 203	7.59	97.83	92.80	95.25
DIRIMPOTS	52 520	56 586	6 099	2 033	50 487	3.59	89.22	96.13	92.55
PLSJUSTICE	81 074	69 559	637	12 152	68 922	17.47	99.08	85.01	91.51
PLSFETES	37 663	35 823	0	1 840	35 823	5.14	100.00	95.11	97.50
PLSRHIN	84 833	74 738	1 026	11 121	73 712	14.88	98.63	86.89	92.39
					**Mean**	**9.06**	**97.35**	**91.75**	**94.35**
					**Median**	**6.64**	**98.86**	**93.69**	**93.90**

**Table 3 sensors-20-02161-t003:** Comparative table showing the quantitative results of the classification for Kasepuhan using Metashape and M_HERACLES.

Class	%Precision		%Recall		%F1 Score	
	Metashape	M_HERACLES	Metashape	M_HERACLES	Metashape	M_HERACLES
Buildings	51.44	95.40	73.49	77.14	60.52	85.30
Walls	6.48	96.61	3.17	77.21	4.26	85.83
Trees	92.15	88.23	85.12	74.80	88.50	80.96
**Median**	***51.44***	***95.40***	***73.49***	***77.14***	***60.52***	***85.30***

**Table 4 sensors-20-02161-t004:** Quantitative analysis on the results of step 2 for the detection and classification of columns in the Kasepuhan dataset. %P is precision, %R is recall, and %F1 is the normalized F1 score.

Object	Point Number		Misclassed		True Positive	%Unclassed	%P	%R	%F1
	Manual	Auto	Overclassed	Unclassed					
K01	2 963	2 106	2 106	0	857	28.92	100.00	71.08	83.09
K02	2 543	1 819	1 815	4	728	28.63	99.78	71.37	83.22
K03	2 577	1 787	1 783	4	794	30.81	99.78	69.19	81.71
K04	2 379	1 618	1 618	0	761	31.99	100.00	68.01	80.96
K05	3 698	2 340	2 340	0	1 358	36.72	100.00	63.28	77.51
K06	3 440	2 158	2 158	0	1 282	37.27	100.00	62.73	77.10
K07	3 646	2 282	2 282	0	1 364	37.41	100.00	62.59	76.99
K08	3 361	2 117	2 117	0	1 244	37.01	100.00	62.99	77.29
					**Mean**	33.60	99.94	66.40	79.73
					**Median**	34.36	100.00	65.64	79.23

**Table 5 sensors-20-02161-t005:** Quantitative analysis on the results of step 2 for the detection and classification of columns in the St-Pierre dataset. %P is precision, %R is recall, and %F1 is the normalized F1 score.

Object	Point Number		Misclassed		True Positive	%Unclassed	%P	%R	%F1
	Manual	Auto	Overclassed	Unclassed					
S01	72 587	54 995	47 709	7 286	24 878	34.27	86.75	65.73	74.79
S02	66 298	64 952	52 922	12 030	13 376	20.18	81.48	79.82	80.64
S03	74 430	55 979	50 435	5 544	23 995	32.24	90.10	67.76	77.35
S04	71 667	59 277	43 647	15 630	28 020	39.10	73.63	60.90	66.67
S05	64 893	54 969	54 343	626	10 550	16.26	98.86	83.74	90.68
S06	66 678	61 804	50 018	11 786	16 660	24.99	80.93	75.01	77.86
S07	67 316	75 062	51 996	23 066	15 320	22.76	69.27	77.24	73.04
S08	60 165	49 212	35 814	13 398	24 351	40.47	72.77	59.53	65.49
					**Mean**	28.78	81.72	71.22	75.81
					**Median**	28.61	81.20	71.39	76.07

**Table 6 sensors-20-02161-t006:** Quantitative analysis on the results of step 2 for the detection and classification of columns in the Valentino dataset. Note that only the detected columns were taken into account here. %P is precision, %R is recall, and %F1 is the normalized F1 score.

Object	Point Number		Misclassed		True Positive	%Unclassed	%P	%R	%F1
	Manual	Auto	Overclassed	Unclassed					
V01	35 370	46 666	15 594	4 298	31 072	12.15	66.58	87.85	75.75
V02	35 845	47 358	15 744	4 231	31 614	11.80	66.76	88.20	75.99
V03	39 169	51 853	17 333	4 649	34 520	11.87	66.57	88.13	75.85
V04	40 155	51 923	17 010	5 242	34 913	13.05	67.24	86.95	75.83
V05	38 288	52 623	17 575	3 240	35 048	8.46	66.60	91.54	77.10
V06	39 689	53 016	17 406	4 079	35 610	10.28	67.17	89.72	76.82
					**Mean**	11.27	66.82	88.73	76.23
					**Median**	11.84	66.68	88.16	75.92

**Table 7 sensors-20-02161-t007:** Quantitative analysis on the results of step 2 for the detection and classification of beams. %P is precision, %R is recall, and %F1 is the normalized F1 score.

Object	Point Number		Misclassed		True Positive	%Unclassed	%P	%R	%F1
	Manual	Auto	Overclassed	Unclassed					
Beam1	15 036	10 960	608	4 684	10 352	42.74	94.45	68.85	79.64
Beam2	57 986	43 826	0	14 160	43 826	32.31	100.00	75.58	86.09
Beam3	28 789	26 141	2 355	5 003	23 786	19.14	90.99	82.62	86.60
					**Mean**	**31.40**	**95.15**	**75.68**	**84.11**
					**Median**	**32.31**	**94.45**	**75.58**	**86.09**

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
