# Peer review of "Virtual Disassembling of Historical Edifices: Experiments and Assessments of an Automatic Approach for Classifying Multi-Scalar Point Clouds into Architectural Elements"

_sensors, 2020, doi:10.3390/s20082161_

Round 1

Reviewer 1 Report

Many photogrammetric methods using imagery and point clouds have been developed and investigated in the field of 3D heritage preservation and documentation. However, the final 3D point clouds only for representation are sometimes not sufficient for analysis, updating, maintenance, and future predictions. Semantic annotation of point cloud data is still a topic of huge research, and advances in this direction, such as automation in labelling are very much welcome. Therefore, the proposed workflow, including the algorithm approach, including automatic functions for semantic segmentation of point cloud datasets, is definitely added scientific value. The presented workflow in the form of a toolbox provides an opportunity to automate most of the 3D processing pipeline reducing time and human intervention. The approach is novel and looks promising not only in the field of heritage documentation but more mapping purposes. It is quite beneficial that the algorithm can annotate automatically GIS attributes to the segmented clusters without spending much time and to perform accurate segmentation and classification. As a further exploration, it is recommendable to use the developed method for the preparation of training data for machine and deep learning algorithms. The paper is very well written and structure. The English language is at a high level. Therefore, I give my strongest recommendation for its publication.

Author Response

The authors would like to thank the reviewer for their remarks. Indeed, the use of our algorithms for the preparation of training data in deep learning algorithms is the subject of a current ongoing experiments.

Reviewer 2 Report

Please check the comments in the attached file. 

Author Response

The attached PDF contains the revised version of the paper with the changes highlighted.

p.1 the final classification results refer to just some particular elements, I would specify that in the title, without talking about "Large Point Cloud Classification"

Response: We agree with the reviewer about the vagueness of this term. In accordance with another remark by Reviewer #3, we have changed the paper title accordingly.

p.1 line 8 as above, be more precise

Response: The authors agree with the reviewer’s comment. We have modified the text to be more precise in our wording.

p.7 line 258 not really clear

Response: We agree that the sentence is not clear enough. We have therefore split the said sentence into two, in hope of clarifying the idea that we wish to convey.

p.8 figure 3 This paragraph needs some English rephrase

Response: Thank you for pulling our attention to this particular text. We have rephrased it to make the meaning clearer.

p.8 line 266 Consider rewriting to avoid some of these words: the, to , in order to, of, at the

Response: As has been pointed out by the reviewer, we have modified the phrase appropriately.

p.8 line 276 try to be consistent with the tenses (use or present or past)

Response: Thank you for this important remark. We generally try to use past tense in describing our experiments.

p.8 line 282 too colloquial language

Response: We agree with the reviewer that the phrase is too colloquial. We have modified it accordingly.

p.11 line 318 I would explain also what you mean with "overclassed" and "unclassed"

Response: The authors also see that it is necessary to explain those two terms. We have added additional explanation to that effect.

p.12 line 334 what do you mean?

Response: We agree with the reviewer that this term is vague. We have modified it to better reflect the idea we wanted to convey.

p.13 line 379 a bullet list with the steps could be helpful to understand better the process.

Response: The authors think that the reviewer’s suggestion is a good idea. We have therefore rewritten the said paragraph to include a bullet list.

p.19 line 559 I think this sentence is a bit too pretentious..

Response: We agree with the reviewer that the sentence may seem too hyperbolic. We have modified the said sentence accordingly.

Reviewer 3 Report

General Comment

This paper illustrates a processing pipeline for segmenting a cloud of 3D points originated by different 3D sensing techniques such as TLS and drone photogrammetry, aimed at documenting historical buildings in their natural landscape.

The implemented tool, called M_HERACLES, is based on the preselection of subsets of the entire 3D point cloud based on a GIS shapefile. On each resulting sub-cloud, a set of algorithms are applied: one taken from the public domain for identifying the ground (Cloth Simulation Filtering). Once the ground data are segmented, three categories of structures are identified with three algorithms: two developed by the authors for identifying supports (i.e., columns) and beams, and one from the public domain (cuboid RANSAC) for identifying beam cuboids.

Although this paper honestly states below the title the following sentence: 

“This paper is an extended version of our paper published in the 8th International Workshop 3D-ARCH, 6-8 February 2019, Bergamo, Italy as well as another paper presented in the 27th International CIPA Symposium, 1-5 September 2019, Ávila, Spain”

this reviewer doesn’t see so many differences with respect of the other two papers already published:

  1. Murtiyoso, A.; Grussenmeyer, P. Point cloud segmentation and semantic annotation aided by GIS data for heritage complexes. The International Archives of the Photogrammetry, Remote Sensing and Spatial Information Sciences, 2019, Vol. XLII, pp. 523–528.
  2. Murtiyoso, A.; Grussenmeyer, P. Automatic Heritage Building Point Cloud Segmentation and Classification Using Geometrical Rules. International Archives of the Photogrammetry, Remote Sensing and Spatial Information Sciences, 2019, Vol. XLII-2/W15, pp. 821–827. doi:10.5194/isprs-archives-xlii-2-w15-821-2019.

Both of them were peer-reviewed papers that already explain this segmentation method extensively, with a significant overlap with the submitted one.

Here there are no improvements in the algorithm that is precisely the same described in the mentioned papers nor a more extensive explanation of the methods.

Also, most of the datasets are the same. Kasepuhan Palace has already been used in the two mentioned papers, as well as part of the St-Pierre dataset. 

For this reason:

  • the left part of figure 4 in the submitted paper is exactly the same in figure 3 in paper 1.
  • Figure 5a of the submitted paper is exactly the same in figure 6 in paper 1.
  • The upper part of the graphical table in figure 6, is partly the same presented by Table 1 in paper 2.

As an “extended version,” one could expect some more analysis, especially in comparing the presented approach with other possible strategies rather than a merge of two previously published papers.

Also, the dataset Kasepuhan Palace has been already used in the two mentioned papers, as well as part of the St-Pierre dataset. Contrarily, the Valentino data are entirely new, as well as the HK Castle.

Therefore, the only new contribution here seems to be the application of the pre-existing algorithm to different datasets estimating more extensively how M_HERACLES works with different types of buildings.

This also raises a concern about the scientific focus suggested by the title: “Virtual Disassembling of Historical Edifices: 3D Pipeline Automation for Large Point Cloud Classification”. Introducing the pipeline is not the topic of this paper because such a presentation has already been done in the two older articles mentioned above.

For this reason, I think that the paper in this form would be just a third repetition of the same article and should be rejected, not adding any innovative scientific contribution in terms of “3D Pipeline Automation for Large Point Cloud Classification”.

On the other hand, if rearranged as a quality characterization of M_HERACLES and re-titled accordingly, it would focus better on its real contribution as a characterization of the method, especially exploring:

  • a quantitative comparison of M_HERACLES with other pre-existing approaches, making clear what is the improvement determined by this tool;
  • the influence of the intrinsic shape files uncertainty when used for slicing the 3D cloud, possibly comparing it with alternative segmentation approaches.

Specific Comments

Page 2, lines 40-41

I would avoid to point out the difference as stated in this sentence. Although widely used in the geomatic area, the association of the word LIDAR with aerial laser scanning is basically wrong, being LIDAR a broader term just indicating a distance measurement obtained by illuminating the target with laser light and measuring the reflected light with a sensor. So any TOF or PS device is a LIDAR sensor independently of the location from where the laser beam is illuminating from! ALS is the right acronym that I would suggest instead. I, therefore, suggest to change the sentence in parenthesis as follows: (including both the Terrestrial Laser Scanning or TLS and the Aerial Laser Scanning or ALS)

Page 2, lines 42-45

This is slightly misleading since this progression in speed makes sense only if you specify that those models are all Time Of Flight (TOF) devices. With Phase Shift TLS, you would have significantly different numbers. I would change the sentence this way: "Using as comparison parameter the scan rate of Time of Flight devices produced for example by Trimble, the point per second rate has improved..."

Author Response

Thank you for the review. We have modified the paper according to the reviewer's comments; the modified version is attached with the changes highlighted.

Here there are no improvements in the algorithm that is precisely the same described in the mentioned papers nor a more extensive explanation of the methods.

Also, most of the datasets are the same. Kasepuhan Palace has already been used in the two mentioned papers, as well as part of the St-Pierre dataset. 

Response: Thank you for this remark. Indeed, the paper was conceived to describe the entirety our proposed workflow which has been separately explained in the two mentioned papers. The Kasepuhan dataset was also indeed used in both of the two previous papers, however the reviewer would notice that in our current paper we used a different part of this dataset for the Step 2. We kept the St-Pierre dataset in Step 2 because we thought that the results regarding the twin pillars were interesting, and since we already stated that the paper is an extended paper we took the liberty of doing so. We also added a new function for the beam detection which was previously unpublished. That being said, we also agree with the reviewer that in this current form the paper’s majority is a repetition of previous papers. We have therefore added a dedicated section on the comparison of our algorithms against other solutions. We have also modified the title to reflect this fact.

For this reason:

the left part of figure 4 in the submitted paper is exactly the same in figure 3 in paper 1.

Figure 5a of the submitted paper is exactly the same in figure 6 in paper 1.

The upper part of the graphical table in figure 6, is partly the same presented by Table 1 in paper 2.

Response: We agree that some of the figure and table elements are identical with previous papers. We have decided to delete Figure 5a. However, we decided to keep the left part of Figure 4 in order to be consistent and balanced in describing both the old Kasepuhan and the new St-Pierre datasets. We have also decided to keep Figure 6 (Figure 10 in the new version) as is, since as we have previously mentioned for the Kasepuhan dataset a different sample was used while St-Pierre is kept for the reasons explained in our response to the previous remark.

As an “extended version,” one could expect some more analysis, especially in comparing the presented approach with other possible strategies rather than a merge of two previously published papers.

Response: We wholeheartedly agree with the reviewer’s propositions. As mentioned before, we have added more analysis by comparing the results against other solutions. Namely, we added a comparison between our Step 1 solution and results from the automatic classification function of the commercial software Agisoft Metashape, with emphasis on the advantages of our algorithms. For Step 2, an additional analysis by comparing the results of Valentino to deep learning results of the same dataset in Malinverni et al. (2019) was performed, although we stress that one of our goals is not to confront our algorithms with DL but rather to support and improve it in creating training data.

This also raises a concern about the scientific focus suggested by the title: “Virtual Disassembling of Historical Edifices: 3D Pipeline Automation for Large Point Cloud Classification”. Introducing the pipeline is not the topic of this paper because such a presentation has already been done in the two older articles mentioned above.

Response:  Thank you for this remark regarding the title. We do agree with the reviewer’s reasoning in this regard. In conjunction with the remarks of Reviewer #2, we have modified the title to better reflect the content of the paper.

Page 2, lines 40-41

I would avoid to point out the difference as stated in this sentence. Although widely used in the geomatic area, the association of the word LIDAR with aerial laser scanning is basically wrong, being LIDAR a broader term just indicating a distance measurement obtained by illuminating the target with laser light and measuring the reflected light with a sensor. So any TOF or PS device is a LIDAR sensor independently of the location from where the laser beam is illuminating from! ALS is the right acronym that I would suggest instead. I, therefore, suggest to change the sentence in parenthesis as follows: (including both the Terrestrial Laser Scanning or TLS and the Aerial Laser Scanning or ALS)

Response: Thank you very much for this important remark. We agree with the reviewer as to this reasoning. We have therefore modified the sentence accordingly.

Page 2, lines 42-45

This is slightly misleading since this progression in speed makes sense only if you specify that those models are all Time Of Flight (TOF) devices. With Phase Shift TLS, you would have significantly different numbers. I would change the sentence this way: "Using as comparison parameter the scan rate of Time of Flight devices produced for example by Trimble, the point per second rate has improved..."

Response: We agree with the reviewer on this matter that the information on the type of TLS is important in this context. We have taken the reviewer’s suggestion and changed the sentence.

Round 2

Reviewer 3 Report

I appreciate the hard work of the authors for improving the paper and its title according to the suggestions.

First of all I think that the scope of this paper is now better focused.

Secondarily, the comparison with other tools makes clear the actual effectiveness of the proposed approach, that is significant.

I think that the paper is now suitable for publication: I just suggest a minor tiny adjustment that I unfortunately missed that in the previous reading, sorry.

As you have agreed the word LIDAR alone should not be used with the implicit meaning of a laser scanner operating from an aircraft. Therefore in the final version I will fix the following occurences of that acronim with either "aerial LIDAR" or "ALS":

Page 4, 166 
Page 10, 316 
Page 12, 349 

Author Response

We thank the reviewer for the time and effort concecrated for our paper, as well as their useful suggestions and comments. We have fixed the terms used from "LIDAR" to either "aerial LIDAR" or "ALS" throughout the paper as suggested by the reviewer.